# Mineral Addition and Mixing Methods Effect on Recycled Aggregate Concrete

**DOI:** 10.3390/ma14040907

**Published:** 2021-02-14

**Authors:** Hasan Dilbas, Mehmet Şamil Güneş

**Affiliations:** 1Department of Civil Engineering, Yuzuncu Yil University, Van 65080, Turkey; 2Department of Statistics, Yildiz Technical University, Istanbul 34220, Turkey; msgunes@yildiz.edu.tr

**Keywords:** recycled aggregate concrete, silica fume, mixing process, compressive strength

## Abstract

This paper presents influence of treatment and mixing methods on recycled aggregate concretes (RAC) designed regarding various techniques. Absolute Volume Method (AVM) according to TS 802, Equivalent Mortar Volume Method (EMV), silica fume (SF) as a mineral addition were considered in the design of concretes. In total, four groups of concretes were produced in the laboratory: (1) natural aggregate concrete (NAC) designed with AVM as control concrete, (2) RAC designed with AVM as control RAC, (3) RAC with SF as a mineral addition designed with AVM as treated RAC and (4) RAC designed with EMV as treated RAC. The tests were performed at 28th days and the statistical analysis were made on the test results. According to the results, EMV and SF increased the compressive strength of concretes and this resulted an increase in the strength class of concrete. A significant statistical difference between the concretes were determined. According to multiple comparison analysis, it was found that especially there was a significant relationship among NAC, RAC and RAC-EMV. In addition, it was recommended that EMV and AVM with 5% SF could be used in the design of RAC rather than AVM only to achieve the target strength class C30/37.

## 1. Introduction

Demolition of concrete structures and waste concrete products have been mainly discussed in the countries. Authorities were worked on identifying a struggle process with the huge mass of waste concrete and offered environmental solutions for the use of waste concrete in newly manufactured concrete as recycled aggregate (RA) taking measures such as regulation and standards for RA [1]. Environmental approaches giving zero harm to the nature, recycling materials, and preserving natural resources so to the economic development have been defined as main aim by countries [2]. However, the authorities faced difficulties to act laws, regulations, and measures on the use of RA in concrete and it is emerged as a need to form a new code for recycled aggregate concrete (RAC) while present concrete codes have been developed for natural aggregate concrete (NAC). On the other hand, RA has attached old mortar (AOM) and natural aggregate (NA) phases and the ambiguity in the properties of RA due to its heterogeneity limits the use of RA in concrete. Hence, absence of reliability does not give hope to mix designer for structural concrete.

To solve the complexity of the use of RA in new concrete safely, some researchers proposed new mixture prescriptions including mineral additions (i.e., silica fume (SF) [3,4], ground granulated blast furnace slag (GGBS) [5,6], metakaolin [7], fly ash (FA) [8,9,10,11], fibers with/without mineral addition (i.e., polypropylene [9], polypropylene + SF [12], steel fiber + SF [4], basalt fiber [13,14], basalt fiber + nano-silica [15], waste-plastic strip + FA [16], steel fiber + FA [17], steel fiber + GGBS [18], glass fiber + FA [19] or redefining mixing method (i.e., Equivalent Mortar Volume Method (EMV) [20,21], two stage mixing method [22]. The suggestions presented satisfactory results and gave confidence to mix designers of RAC. However, despite the diversity of the suggestions, the most useful technique is not clear and comparison of them is required. Thus, the designer should compare the useful ones. In this perspective, Mineral Addition Treatment (MAT) and Equivalent Mortar Volume Methods (EMV) were the well-known and widely considered methods in the literature [3,20,23].

In this experimental research, the properties of concretes (NAC, RAC, RAC included SF as a mineral addition and RAC designed with EMV) were compared to observe the effect of treatment methods (Figure 1). Here, NAC, RAC and RAC with silica fume were designed with AVM. 120 concrete specimens for four group of concrete were produced in the laboratory. Then, 28th day compressive strength of concretes was determined, and the statistical analysis were conducted.

## 2. Materials and Methods

### 2.1. Materials

General purpose CEM I cement suitable with TS EN 197-1 [24] was used in the concrete mixes. The properties of cement and silica fume (SF) are given in Table 1.

In the concrete mixes, natural coarse aggregate and recycled coarse aggregate were used as the coarse ones and the granulometry of the mixes are the same. Natural gravel was crushed, calcareous aggregate and also sand was utilized as fine aggregate in the mixes (Table 2). Super plasticizer was used to enhance the low workability of fresh concretes (Table 3). The slump class is set to S2 for all mixes [25].

### 2.2. Concrete Design Method and Data Evaluation Approachs

Four concrete mixes were produced in the laboratory with the target strength class C30/37 (Table 4). Absolute Volume Method (AVM) and Equivalent Mortar Volume Method (EMV) were considered to design the mixes (Table 5). According to AVM a unit volume (it is generally 1 m^3^) of concrete is filled with the components of concrete (Equation (1)) [23]:(1)$$V_{1m}^{3}=V_{agg}+V_{cem}+V_{w}+V_{ch}+V_{air}$$

Here, *V_agg_* is volume of aggregate, *V_cem_* is volume of cement, *V_w_* is volume of water, *V_ch_* is volume of chemicals and *V_air_* is volume of air in concrete.

EMV requires that recycled aggregate concrete has same amount of total mortar volume with control concrete so to constant aggregate volume. Hence the residual content should be determined for RA [20]. HCl solution can be used to determine the amount of residual on RA [26] (Table 2). The remaining part over 4mm sieve was determined in the residual defining test after 0.1 M HCl solution attack to RA in a container. EMV requires the constant volume of aggregate as (Equation (3)) [21]:

EMV requires the constant volume of aggregate as (Equation (3)) [20]:(2)$$V_{RCA}^{RAC}=\frac{V_{NA}^{NAC}x\left(1-R\right)}{\left(1-RMC\right)x\frac{SG_{b}^{RCA}}{SG_{b}^{OVA}}}$$

Here, $V_{RCA}^{RAC}$ is the volume ratio of coarse in RAC, $V_{NA}^{NAC}$ is the volume ratio of fresh natural aggregate in control concrete, $SG_{b}^{RCA}$ and $SG_{b}^{OVA}$ are the bulk specific gravity of RA and original virgin aggregate, respectively, *RMC* is the residual mortar content of RA and *R* is the volume fraction of fresh natural aggregate content of RAC to fresh natural aggregate content of control mix.

The cement quantity and water-to-cement ratio was kept constant for all concrete mixes. Concrete was cast incompatible with ASTM C192/C192M–13a [27] and vibration was applied on the fresh concrete. For each concrete group, 30 cube specimens (15 × 15 × 15 cm) were produced and cured in lime saturated water for 28 days. At the end of the time (28th days), 120 concrete specimens were tested in 3000kN compression machine in accordance with TS EN 12390-3 [28] and the results are given in Table 5.

#### 2.2.1. Strength Class Determination

95% confidence interval was considered, and strength class of concrete groups were determined using Equations (3) and (4) [25]:*f*_*c*,*avg*_ ≥ *f*_*ck*_ + 1.96 *σ*(3)
*f*_*c*,*min*_ ≥ *f*_*ck*_ − 4.0(4)

Here, *f_ck_* characteristic compressive strength of group (MPa), *f_c_, _avg_* is the average compressive strength of group (MPa), *σ* standard deviation, and *f_c_, _min_* is the minimum compressive strength of group (MPa).

#### 2.2.2. Statistical Analysis Method

In this study, Shapiro-Wilk normality test was performed on NAC, RAC, RAC-SF, RAC-EMV [29]. The Pearson correlation coefficient was used to test the degree of relationship between concrete types and to obtain information about the general structure of the results. Afterwards, the variance analysis (ANOVA) was used to measure whether the compressive strength values had a significant effect on the concrete types at 5% significance level. Also, the Games-Howell multiple comparison test was used to measure whether there was a significant difference between the concrete types of compressive strengths (where group variances were not equal). The analysis was performed using IBM SPSS 22 at 5% significance level.

## 3. Results

### 3.1. Strength Class of Concretes

According to the results given in Table 6, the target strength class C30/37 was achieved for NAC and RAC-SF. The strength class of RAC-EMV was found as C35/45 and however, it was found as C25/30 for RAC. Poor properties of RA influenced the concrete properties and decreased the compressive strength of RAC [3,30,31,32,33,34,35]. Attached old mortar (AOM) content in RA had an important role on the decrease of compressive strength and AOM had porous structure with lower strength characteristics [36]. However, silica fume (SF) use in concrete mix increased the compressive strength and also the strength class of concrete giving satisfactory results. Here, SF showed two significant behaviors: 1) Causing extra C-S-H gels in matrix bounding free Ca(OH)_2_ in the cement paste, 2) Filler effect (closing concrete pores) [37]. In addition, EMV, also, gave a comparable result to RAC-EMV and caused an increase in the compressive strength and the strength class [20]. This success was sourced by aggregate concentration consideration in the mix. Besides, the similar findings with the current literature are achieved observing the lower compressive strength and higher standard deviation values of compressive strength compared to control ones [3,38,39,40,41,42,43].

### 3.2. Comparison of the Methods

As given in Table 6, the control concrete (NAC) that was designed with AVM had C30/37 strength class and the consideration of EMV as a mixing approach in the production of concrete ensured C35/45 but also C30/37 (the upper strength class covers and ensures the lower ones). Besides, SF treatment gave approximately close compressive strength and strength class with control concrete (NAC). However, increase in the strength class of RAC from C25/30, which is for RAC, to C35/45 which is for RAC-EMV, due to the consideration of EMV was not similar with the increase in the strength class of RAC from C25/30 which is for RAC to C30/37 which is for RAC-SF due to the consideration of SF. Although EMV seemed to be a potential to increase the strength class of RAC, EMV increased the standard deviation values of RAC-EMV. On the other hand, it was clear that SF decreased the standard deviation of the test results and the minimum standard deviation was calculated for RAC-SF and RA marginally changed the standard deviation values of RAC [3,33]. More tests should be conducted to observe the exact behavior of RAC.

### 3.3. Statistical Results

In the Table 7, the statistical values of concretes are given as standard deviation, mean, standard error and 95% confidence interval with histograms. According to Table 7, the lower bound of NAC crossed with upper bound of RAC-SF although the means of NAC and RAC-SF were different. Besides, the lower and upper bounds of RAC and RAC-EMV did not cross with the lower and upper bounds of NAC.

Shapiro-Wilk normality, skewness, kurtosis ratio to standard error results are given in Table 8. The values obtained from the standard error division of the observed kurtosis and skewness values for all variables varies between (−2, 2) indicating that the data was distributed normally. Furthermore, the Shapiro-Wilk normality statistics gave significant results for all qualifications at 5% significance level.

The Pearson Correlation coefficient was a measure of the variation of two or more variables. The conducted correlation analysis showed how a change in interrelated variables affected the other variables and the relationship among them. For the correlation coefficient (it takes values between −1 and 1), ”0” is the non-correlation, ”−1” represents the perfect negative relationship and,”1” represents the perfect positive relationship. The interpretation of the correlation coefficient is given in the Figure 2. This figure showed samples of what vary correlations remind, in terms of the strength and direction of the relationship with histograms and fit lines. Pearson correlation coefficient between NAC and RAC was equal −0.20 which showed that there was a very weak relation between those variables. On the other hand, Pearson correlation coefficient between NAC and RAC-SF was equal 0.20 and that of NAC and RAC-EMV was equal −0.059.

A scatterplot gives the relations between two variables measured for the dataset. Each individual in the data appears as a point on the graph. It is convenient to use scatter plots with correlation test results. As shown in Figure 2, correlation coefficient between NAC and RAC was equal −0.20. As it could be seen from the scatter plot in Figure 3, relationship between NAC and RAC had a negative line (downhill) with confidence interval (CI) which indicated the same interpretation with correlation coefficient.

Variance analysis (ANOVA) measures the significance of compressive strengths on concrete types and the results of ANOVA is given in Table 9. Statistically, the relationship between the groups were significant at %5 significance level (*p*-value = 0.00 < 0.05). It is known that TS-500 [42] considered %10 significance level with higher tolerance compared to 5% significance level [43]. Also, a multiple comparison test was used to determine which concrete types were significant (Table 10) and here, if the confidence intervals for the multiple comparison test contained a value of “0”, the bilateral relationship was not meaningful. Accordingly, when Table 10 was examined, it was found that the relationships of (NAC)-(RAC), (NAC)-(RAC-EMV), (RAC-SF)-(RAC), (RAC-EMV)-(RAC) and (RAC-EMV)-(RAC-SF) were significant at 5% significance level. However, the relationship between (NAC)-(RAC-SF) was not significant.

## 4. Conclusions and Discussions

In this paper, a statistical study was conducted, and the compressive strength test results of concretes designed with Absolute Volume Method (AVM) and Equivalent Mortar Volume Method (EMV) and included natural aggregate (NA), recycled aggregate (RA) and silica fume (SF) were investigated. Based on the results, the following conclusions were made:Mineral Addition Treatment Method with SF and EMV gives convincing results eliminating the negative effect of attached old mortar (AOM) in RA.The target strength class C30/37 is achieved for NAC, RAC-SF and RAC-EMV, and however, RAC cannot achieve C30/37. SF addition and EMV facilitate to obtain the target strength class.The compressive strength of all concretes distributes normally according to the Shapiro-Wilk normality test and Skewness and Kurtosis to standard error ratios.It is found that the correlation analysis and the scatter plots give compatible results. The correlation analysis and the scatter plots indicate that the relation between concretes behavior pattern is observed at low level. Especially, for instance, the relation between behaviors of NAC and RAC is equal −0.20 and it means that there is very weak relation due to RA.According to the results of ANOVA, the relationship between the concretes are significant at 95% reliability level, although generally concrete standards consider 90% reliability level.Games-Howell Multiple Comparison Test demonstrates that the most significant bilateral relationships between (RAC-EMV)-(RAC) and (RAC)-(NAC) are found. (It is noted that the correlation test measures whether there is a relation between variables while ANOVA and comparison tests measure significance grades. These evaluation approaches should not be confused.)

In summary, according to the statistical evolutions, there is a major difference between the concretes and this phenomenon depends on the utilized components and considered mixing methods generally. However, Mineral Addition Treatment Method (here it is SF) and EMV are useful to improve the performance of RAC, and especially, EMV is strongly recommended for RAC mix design by the authors instead of AVM. If AVM is considered, SF addition use in mixes is recommended by the authors.

In addition, the following discussions were made after the evaluation of the results and conclusions:The data used in the statistical approaches were collected from experiments and at first strength class of concrete series were determined. In this point it was thought whether the concretes were in the required strength class (C30/37), and the results were checked in consideration of the helpful evaluation equations given in the related codes. However, it is well-known that the concrete, commonly used in the engineering area, includes the natural aggregate and, also the compressive strength results of natural aggregate concrete are distributed compatible with the normal distribution function. Here, it is expected that the concretes included recycled aggregate and designed with different mixing methods would show a similar behavior with natural aggregate concrete. However, the truth of recycled aggregate concrete was different. For instance, as a result of the normality test evaluations (skewness, kurtosis, etc.), when recycled aggregate was considered, the distribution of the test results of recycled aggregate concrete presented a non-similarity with natural ones although the use of silica fume changed a bit the behavior of recycled aggregate concrete from recycled aggregate concrete to natural aggregate concrete. According to this, it could be concluded that despite the consideration of silica fume in recycled aggregate concrete, natural aggregate concrete and recycled aggregate concrete had different characteristics and it was thought that the observed difference depended on the components such as recycled aggregate. To present the difference/similarity of concretes behavior, in addition, the comparison techniques were employed and hence the difference between the concrete types was obviously seen. The first comparison technique was made in consideration of Person Correlation Coefficient and the most suitable similarity between recycled aggregate concrete and natural aggregate concrete was found as 0.20 (the higher is good up to 1.0 and down to −1.0). The second comparison technique was made in consideration of ANOVA with Games-Howell Multiple Comparison Test and Games-Howell Comparison Test had an interrelation assessment approach. As expected, the first and the second approaches demonstrated the similar results: There was a specific difference between the concretes included natural and recycled aggregates in dependent of mixing methods such as AVM and EMV and, also mineral addition such as silica fume.The critics and discussions on the results canalized the authors to think that the evaluation of test results of different types of concretes (i.e., natural aggregate concrete, heavy concrete, geopolymer concrete, recycled aggregate concrete) could mislead decision makers and the evaluation of test results of different concrete types may be separated in the standards and the evaluation equations for each concrete type could be proposed in the codes after several trial-and-error tests.Considering the various studies in the literature, Tukey’s Test (it is a comparison test) was mostly used together with ANOVA analysis (i.e., Refs. [40,44]). In general, the crucial and inadequate points in the literature are the assumptions of Tukey’s Test which was not properly considered for the situations examined in the study and, the lack of explanations of Multiple Comparison Tests (i.e., Refs. [45,46]). Therefore, the data were properly examined and discussed in detail in consideration of many statistical approaches in the current paper and one of them was Games-Howell Test. In the test, the assumption is made as there is no equal variance that is provided in the determination of the relationships between the groups. In addition, the application of the test on the recycled aggregate concretes designed with many methods was one of the novelty parts of the study and it clearly ensured the difference of the concretes revealing the characteristics of concretes.

## 5. Future Aspects

Also, there is a lack of knowledge in RAC application included many mineral additions (metakaolin, fly ash, granulated blast furnace slag, etc.) and fibers (basalt fiber, polypropylene fiber, steel fiber, etc.) and also those under the different curing conditions. In this paper only SF as a mineral addition, AVM and EMV as a design method is considered and compared evaluating statistical data obtained various stochastic approaches. In addition, it is though that aggregate types (light, normal and heavy ones), water-to-binder ratio, chemical admixtures, etc. are the other components effecting the statistical results and the approaches considered in this paper can be applied to those. Besides, in the future studies, clustering methods, the analysis of discriminant and regression and multivariate statistical methods etc. are able to be utilized to evaluate the effect of materials, methods etc. on concretes’ relations and behaviors.

## Figures and Tables

**Figure 1 materials-14-00907-f001:**
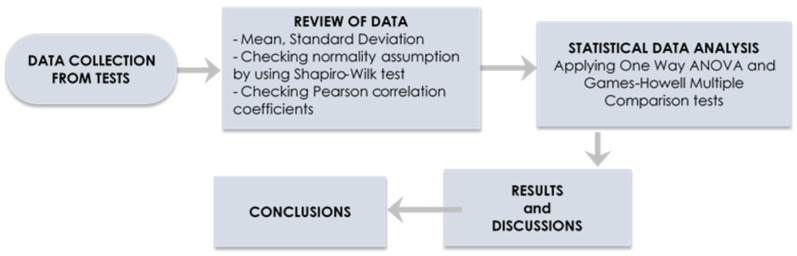
Methodology flow chart.

**Figure 2 materials-14-00907-f002:**
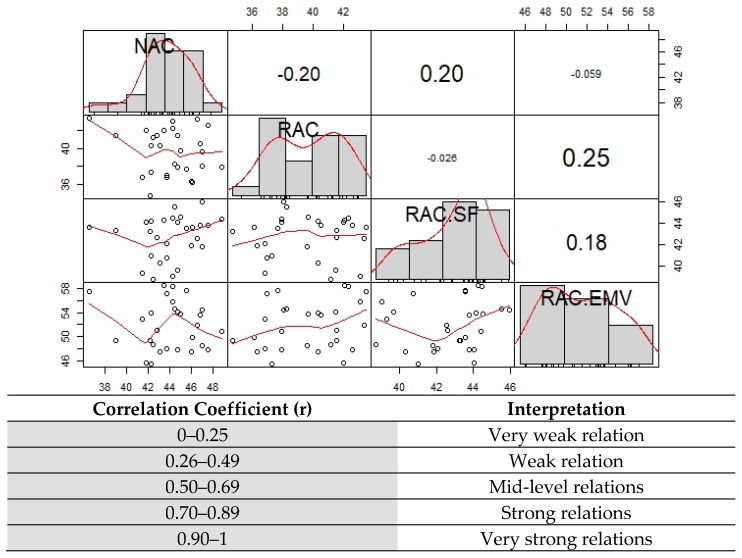
The Pearson Correlation coefficient results.

**Figure 3 materials-14-00907-f003:**
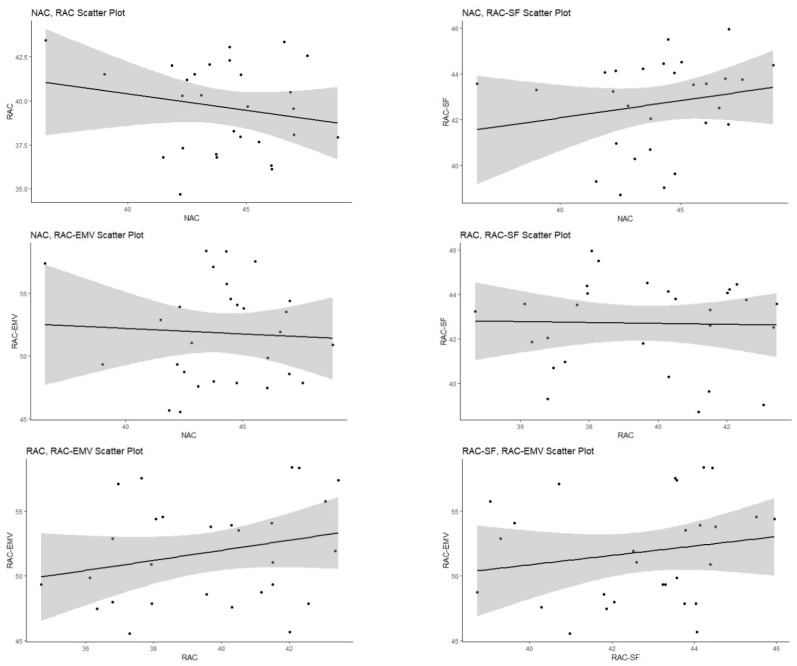
Scatter plots between concrete types with confidence intervals.

**Table 1 materials-14-00907-t001:** Properties of cement and silica fume.

Contents	Cement	SF
SiO_2_ (%)	18.9	91.42
CaO (%)	64.7	0.52
SO_3_ (%)	3.42	0.37
Al_2_O_3_ (%)	4.8	0.72
Fe_2_O_3_ (%)	3.4	1.66
MgO (%)	1.4	0.92
K_2_O (%)	0.4	1.21
Na_2_O (%)	0.7	0.38
Density (g/cm^3^)	3.11	0.642
Chlorine ratio (%)	0.0241	0.04
Specific surface area (m^2^/kg)	3840	21290
Loss on ignition (%)	1.82	1.72
Activity index (%)	-	118

**Table 2 materials-14-00907-t002:** The properties of natural aggregates.

Notation	Density, g/cm^3^	Water Absorption, %	LA Abrasion value, %	Residual Content, %
Sand	2.81	1.31	-	-
NA (11.2–22.4 mm)	2.70	0.75	24	-
NA (4–11.2 mm)	2.73	0.72	-	-
RA (11.2–22.4 mm)	2.00	8.95	55	52.5
RA (4–11.2 mm)	2.06	8.80	-	39.2

**Table 3 materials-14-00907-t003:** The properties of super plasticizer.

Content	Super Plasticizer
Structure of material	Polycarboxylic ether
Color	Amber
Density (kg/l)	1.08–1.14
Alkaline ratio (%)	<3
Chlorine ratio (%)	<0.1

**Table 4 materials-14-00907-t004:** Ingredients of mixes.

Components	NAC	RAC	RAC-SF	RAC-EMV
Cement, kg/m^3^	340	340	323	255
Silica fume, kg/m^3^	-	-	17	-
Water, kg/m^3^	163	163	163	123
Super plasticizer, %	0.75	0.85	0.95	1.55
Sand, kg/m^3^	806	806	806	608
Aggregate 4–11.2 mm, kg/m^3^	392	-	-	-
Recycled aggregate 4–11.2 mm, kg/m^3^	-	296	296	368
Aggregate 11–22.4 mm, kg/m^3^	775	-	-	-
Recycled aggregate 11–22.4 mm, kg/m^3^	-	574	574	774

**Table 5 materials-14-00907-t005:** Compression test results of specimens.

Number of Specimen	NAC	RAC	RAC-SF	RAC-EMV
1	36.73	31.05	34.20	48.00
2	35.98	34.87	35.78	42.89
3	35.70	34.59	32.53	40.94
4	35.16	35.29	37.01	38.37
5	35.56	31.33	34.42	38.26
6	38.07	23.83	36.32	42.73
7	37.59	31.88	36.99	40.21
8	35.46	29.14	36.32	41.43
9	38.25	31.63	36.57	48.35
10	37.20	35.52	37.33	49.02
11	39.15	36.41	35.72	43.64
12	37.84	33.33	37.40	45.20
13	36.76	30.91	35.33	40.30
14	39.96	35.75	36.76	40.19
15	37.22	36.18	32.80	46.84
16	41.04	31.86	37.29	42.74
17	39.36	34.02	36.78	44.97
18	36.20	33.86	33.84	39.98
19	38.71	30.34	36.60	41.89
20	30.70	36.50	36.60	48.22
21	39.47	33.24	35.12	40.81
22	36.78	31.83	28.96	45.62
23	38.69	30.52	35.17	39.86
24	35.54	33.84	37.08	45.29
25	39.50	31.98	38.60	45.71
26	32.78	34.87	36.37	41.45
27	36.50	35.34	37.15	49.05
28	37.61	34.84	33.30	45.43
29	34.85	30.90	33.01	44.44
30	37.36	32.15	38.23	45.82

**Table 6 materials-14-00907-t006:** Compressive strength and strength class of concretes.

Parameters	NAC	RAC	RAC-SF	RAC-EMV
Average compressive strength, MPa	44.12	39.20	42.44	51.89
Minimum compressive strength, MPa	36.55	28.37	34.48	45.55
Std. deviation of compressive strength, MPa	2.63	2.52	1.49	3.95
Strength class of concrete	C30/37	C25/30	C30/37	C35/45

**Table 7 materials-14-00907-t007:** Statistical parameters for concretes (with histograms).

Concretes	Mean	Std. Deviation		Std. Error	95% Confidence Interval	
					Lower Bound	Upper Bound
NAC	44.08	2.63		0.49	43.06	45.10
RAC	39.63	2.52		0.47	38.65	40.61
RAC-SF	42.70	1.97		0.37	41.93	43.46
RAC-EMV	51.84	3.95		0.74	50.30	53.37
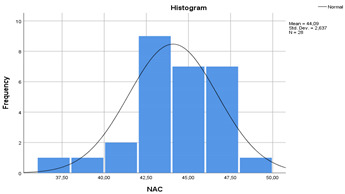 (NAC)			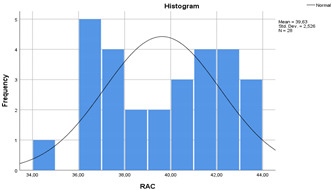 (RAC)			
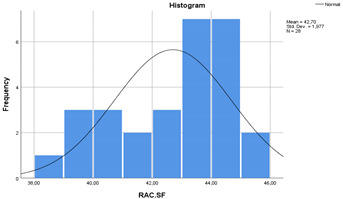 (RAC-SF)			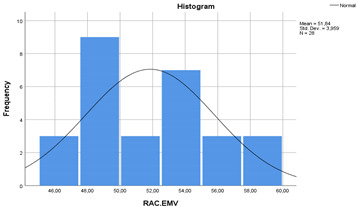 (RAC-EMV)			

**Table 8 materials-14-00907-t008:** Shapiro-Wilk normality test results.

Evaluations	CONCRETES			
	NAC	RAC	RAC-SF	RAC-EMV
Skewness to std. Error ratio	−1.73	−0.31	−1.39	0.38
Kurtosis to std. Error ratio	1.59	−1.41	−0.64	−1.38
Shapiro-Wilk p-value	0.302	0.154	0.058	0.119

**Table 9 materials-14-00907-t009:** ANOVA results with means and interval plots for concretes.

Interactions	Sum of Squares	df	Mean Square	F	Sig.
Between Groups	2267.729	3	755.910	91.849	0.000
Within Groups	888.833	108	8.230		
Total	3156.561	111			
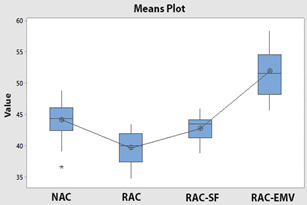			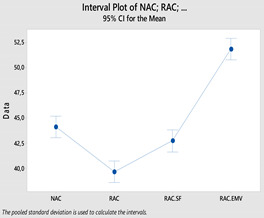		

**Table 10 materials-14-00907-t010:** Games-Howell Multiple Comparison test results and test graph.

(I) kod	(J) kod	Mean Difference (I–J)	Std. Error	Sig.	95% Confidence Interval	
					Lower Bound	Upper Bound
NAC	RAC	40.45 *	0.69	**0.000**	20.62	60.28
	RAC-SF	10.38	0.62	0.131	−00.27	30.03
	RAC-EMV	−70.75 *	0.89	**0.000**	−100.15	−50.36
RAC	NAC	−40.45 *	0.69	**0.000**	−60.28	−20.62
	RAC-SF	−30.06 *	0.60	**0.000**	−40.67	−10.45
	RAC-EMV	−120.20 *	0.88	**0.000**	−140.57	−90.84
RAC-SF	NAC	−10.38	0.62	0.131	−30.03	00.27
	RAC	30.06 *	0.60	**0.000**	10.45	40.67
	RAC-EMV	−90.14 *	0.83	**0.000**	−110.38	−60.89
RAC-EMV	NAC	70.75 *	0.89	**0.000**	50.36	100.15
	RAC	120.20 *	0.88	**0.000**	90.84	140.57
	RAC-SF	90.14 *	0.83	**0.000**	60.89	110.38
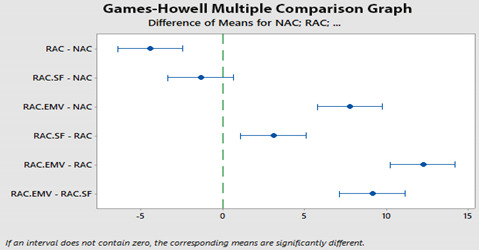 * The mean difference is significant at the 0.05 level.						

## Data Availability

The data and code related to the analysis presented in this paper will be made available upon request.

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
