# Peer review of "Mineral Addition and Mixing Methods Effect on Recycled Aggregate Concrete"

_materials, 2021, doi:10.3390/ma14040907_

Round 1

Reviewer 1 Report

Brief Summary

The manuscript presents an experimental campaign in order to determine the effect of mixing methods and mineral addition on RAC. A non traditional mixing method, Equivalent Mortar Volume Method (EMV) is used and Silica Fume (SF) as mineral addition. Those two are compared against control samples using traditional mixing, Absolute Volume Method (AVM) and no mineral addition. Through a rigorous statistical analysis, results show that EMV and AVM with 5% of SF can be used in the design of RAC with a target strength class C30/37.

Broad Comments

Why doesn’t the study include the control group: NAC with SF and AVM? this would allow to quantify the effect that SF has on RAC compared to that on NAC.

The experimental campaign and results presented are similar to an existing reference in the literature [1], how to justify the originality and novelty of the present contribution?

Specific Comments (line by line)

Half document uses EVM acronym for Equivalent Mortar Volume and the other half uses EMV. Please unify it into one denomination.

Line 66: crushed,

Line 67: ‘is used’, present tense is used here while past tense ‘was utilized’ in the previous line, unify the text in all the document

Line 164: did not cross

Line 228: recycled aggregate (RA)

References

[1] Anike, E.E., Saidani, M., Ganjian, E. et al. Evaluation of conventional and equivalent mortar volume mix design methods for recycled aggregate concrete. Mater Struct 53, 22 (2020). https://doi.org/10.1617/s11527-020-1457-3

Author Response

Dear Editor,

Thank you for giving us the opportunity to the revision of our manuscript titled Mineral Addition and Mixing Methods Effect on Recycled Aggregate Concrete to Materials. We appreciate the time and effort that you and the reviewers have dedicated to providing your valuable feedback on our manuscript. We are grateful to the reviewers for their insightful comments on the paper. We have been able to incorporate changes to reflect most of the suggestions provided by the reviewers. Thank you for encouraging us to submit our revised paper.

We hope the manuscript after careful revisions meet your high standards. The authors welcome further constructive comments if any.

Sincerely

Reviewer 2 Report

The article under review is not well prepared, reading does not flow, some parts of the text need to be read twice for proper understanding.

Sections of the article are not well prepared, f.e. Materials and methods presents fragments of the results (table 5); there is no discussion of the results: what are the new findings of the experiments? how does it compare with the findings of other researches in the field?

What was the quantity of the probe, how many samples has been tested?

What are the limitations of the presented findings?

L. 19-20 "In addition, it was recommended that EVM and AVM with 5%  SF could be used in the design of RAC rather than AVM to achieve the target strength class C30/37." Please explain, should it be: "In addition, it was recommended that EVM and AVM with 5% 19 SF could be used in the design of RAC rather than AVM only to achieve the target strength class C30/37"?.

L88-90: "EVM requires the constant volume of aggregate as (Eq.3) [21]: 89
EVM requires the constant volume of aggregate as (Eq.3) [14]:" There is a mistake in references and in the text.

L. 94: V RCA RCA, there is a syntax error.

L. 135: "RAC-EMV was in C35/45 and however, it was C25/30 for RAC."

L. 153-156: "However, the observed higher increase in the strength class of RAC-EMV due to EMV could not observed for RAC-SF due to SF in comparison to NAC and the target strength class C30/37 was obtained only for RAC-SF."

etc. Many fragments of text is writtten in the way that is hard to understand. Article should be improved in style, form and grammary of used language.

Tables are formatted with different decimal spacers.

Author Response

(The authors gave the same response as above.)

Round 2

Reviewer 1 Report

Brief Summary

The manuscript presents an experimental campaign in order to determine the effect of mixing methods and mineral addition on RAC. A non-traditional mixing method, Equivalent Mortar Volume Method (EMV) is used and Silica Fume (SF) as mineral addition. Those two are compared against control samples using traditional mixing, Absolute Volume Method (AVM) and no mineral addition. Through a rigorous statistical analysis, results show that EMV and AVM with 5% of SF can be used in the design of RAC with a target strength class C30/37.

Broad Comments

The reviewer thanks the authors for the reply letter and the modifications.

The authors made clear through the reply letter what are the contributions of the present paper with respect to the existing literature. The statistical analysis performed gives to the publication originality and overall merit. Modifications according to Reviewer #2 remarks also contribute to improve the quality of the document.

Author Response

Dear Editor,

Thank you for giving us the opportunity to revision (reference to materials-1095443) of our manuscript titled Mineral Addition and Mixing Methods Effect on Recycled Aggregate Concrete to Materials. We appreciate the time and effort that you and the reviewers have dedicated to providing your valuable feedback on our manuscript. We are grateful to the reviewers for their insightful comments on paper. We have been able to incorporate changes to reflect most of the suggestions provided by the reviewers. Thank you for encouraging us to submit our revised paper.

We hope the manuscript after careful revisions meet your high standards. The authors welcome further constructive comments if any.  

Thank you again for your time and effort, and for helping us improve the manuscript.

                                                                                                                               Sincerely

                                                                 Hasan Dilbas and Mehmet Şamil Güneş

Reviewer 2 Report

Most of my concerns has been corrected and explained, thank you.

I would suggest the authors implement the full IMRAD structure (there is a lack of discussion right now). The discussion should implement f.e.  received principles, show what is the meaning and importance of received results, what is the next research plan, ... Part of the discussion of the results is implemented in the results section, but the article would be clearer if a result discussion section was introduced.

Yours Sincerely,

Reviewer.

Author Response

(The authors gave the same response as above.)
